# Interactions of Fusarium Crown Rot of Wheat with Nitrogen

**DOI:** 10.3390/plants12030533

**Published:** 2023-01-24

**Authors:** Mitchell Buster, Steven Simpfendorfer, Christopher Guppy, Mike Sissons, Richard J. Flavel

**Affiliations:** 1New South Wales Department of Primary Industries, Tamworth, NSW 2340, Australia; 2School of Environmental and Rural Science, University of New England, Armidale, NSW 2351, Australia

**Keywords:** nitrogen transfer efficiency, cereal, disease management, *Fusarium pseudograminearum*

## Abstract

The cereal disease Fusarium crown rot (FCR), caused by the fungal pathogen *Fusarium pseudograminearum* (*Fp*), is a major constraint to cereal production worldwide. Nitrogen (N) fertilizer is estimated to be approximately 30% of the input costs for grain growers in Australia and is the primary driver of yield and grain protein levels. When targeting high yield and protein, generous nitrogen fertilizer applications are thought to result in large biomass production, which exacerbates FCR severity, reducing grain yield and quality. This research was undertaken to investigate the effect of temporal N availability in high-protein bread and durum wheat varieties on FCR severity. Laboratory and controlled environment experiments assessed the relationship between FCR and N at a mechanistic and plant level. An in vitro study demonstrated an increase in *Fp* mycelial growth under increased N availability, especially when N was supplied as urea compared with ammonium nitrate. Similarly, under controlled environmental conditions, increased soil N availability promoted FCR severity within infected plants. Stem N transfer efficiency was significantly decreased under FCR infection in both bread and durum wheat varieties by 4.5% and 10.2%, respectively. This new research demonstrates that FCR not only decreases yield and grain quality but appears to have previously unrecognised detrimental impacts on nitrogen-use efficiency in wheat. This indicates that the current impact of losses from FCR may also decrease N-use inefficiencies, as well as yield and quality penalties. An improved understanding of the interactions and restrictions of FCR infection may allow growers to better manage the disease through manipulation of the soil’s temporal N availability.

## 1. Introduction

Wheat is the world’s most important food source, providing 19% of the daily calories and 21% of protein requirements for humans [1,2]. Fusarium crown rot (FCR) of wheat, caused by the fungus *Fusarium pseudograminearum* (*Fp*), is a major cereal disease responsible for yield and quality downgrades around the world [3,4,5]. The disease restricts the ability of wheat plants to transfer solutes and water from roots to shoots, causing significant productivity loss [5,6,7]. FCR has been estimated to cost the Australian wheat industry on average a total of 88 million AUD in yield and quality losses annually [8]. In Australia, the adoption of reduced tillage and stubble retention cropping systems to conserve soil moisture has seen an increase in the prevalence of FCR and associated research concerning the management of this disease [9,10]. An average of 18% of global nitrogen (N) fertiliser is used on wheat crops each year with increasing demand worldwide [11]. N fertiliser is estimated to be approximately 30% of the input costs for grain growers in Australia and is the primary driver of yield and grain protein levels [12]. It is hypothesised that increased rates of N fertiliser application may exacerbate the severity of FCR [13,14]. This poses a conundrum to the production of high-protein bread and durum wheat, where larger N inputs are required to achieve high grain protein content (GPC) grades, for example, at 13–14% in Australia [15]. Plants with high N availability during early growth may deplete the soil water profile quickly due to elevated leaf production during the vegetative phase of wheat growth [13]. This can have a secondary effect on wheat plants infected with FCR by hastening the onset of water-limited stress, which exacerbates disease expression and yield loss. This is especially important in production areas with drier, semi-arid climates, such as the northern grain’s region (NGR) of New South Wales (NSW), Australia, where dry and hot finishes to the wheat growing season are common and nitrogen application is usually supplied at planting [16]. It is important to match water and N supply with peak crop demand [17,18]. However, the primary mechanistic relationship between *Fp* and N availability has not been determined. This study sought to understand both the primary mechanistic relationship and the secondary agronomic effects between temporal availability of N and FCR severity with a view to identifying potential opportunities to improve the management of this disease.

## 2. Results

### 2.1. In Vitro Agar Experiment

The three *Fp* isolates did not significantly vary in their rate of in vitro growth across N treatments (*p* = 0.128); hence, aggregate data for the three *Fp* isolates is presented. Urea had a positive effect on the in vitro growth of *Fp* in the two lowest concentrations compared to the control when measured on days three and six (Figure 1). Only the lowest concentration of urea significantly increased the in vitro growth of *Fp* at day eight. The highest rate of urea had no significant effect on the in vitro growth of *Fp* on day three and had reduced growth on days six and eight compared with the control (Figure 1). Ammonium nitrate at the lowest (0.002 mg/kg) concentration had no significant effect on *Fp* in vitro growth on day three but significantly reduced growth at all other concentrations and time points compared with the control (Figure 1). 

### 2.2. Controlled Environment Experiment

No statistical difference was observed in visual browning severity between the bread and durum wheat cultivars (*p* = 0.2035). FCR severity was increased with *Fp* inoculation in both banded and surface treatments by 10 % and 30 %, respectively, with very low infection levels in the uninoculated treatments (*p* < 0.0001) (Figure 2). Fusarium crown rot severity at GS39 was increased in the surface-applied N treatment by 22% compared to the banded N treatment in the *Fp* inoculated treatment (*p* = 0.003) (Figure 2). Significant differences in early N uptake were not recorded due to sampling occurring too late in the season at GS39. However, there was a non-significant increase in the mean ^15^N% recovered in the surface treatment by 6.5% across both varieties from 2.14% to 2.28% (*p* = 0.202) (data not shown).

In the uninoculated treatment, FCR severity at harvest increased in the banded N treatment by 12.7% in the bread wheat variety LRPB Lancer and 45.0% in the durum variety DBA Lillaroi compared with the surface N treatment (*p* < 0.05) (Figure 3). Differences between N placement treatments were not significant in the inoculated treatment in both varieties (Figure 3).

*Fp* inoculation reduced sNTE on average by 4.5% in LRPB Lancer and 10.2% in DBA Lillaroi (*p* < 0.001) (Figure 4). LRPB Lancer had a 26% higher sNTE compared to DBA Lillaroi in both inoculated and uninoculated treatments (*p* < 0.001) (Figure 4).

In the bread wheat variety, LRPB Lancer *Fp* inoculation reduced sNTE by 9.1% in the banded N treatment (*p* < 0.05) but was not significantly different in the surface N treatment (*p* > 0.05) (Figure 5). In the durum variety, DBA Lillaroi *Fp* inoculation reduced sNTE by 13.8% in the surface N treatment (*p* < 0.05), whilst the reduction was not significant in the banded N treatment. The bread wheat variety LRPB Lancer had a significantly higher sNTE compared with the durum variety DBA Lillaroi under both N and *Fp* inoculation treatments (*p* < 0.001).

The earliest sampling at GS39 demonstrated no significant difference between banded and surface N treatments in tissue N concentration (*p* > 0.05) (Figure 6). Samplings at GS45 and in mature grain demonstrated increased tissue N concentration in the banded treatment compared to the surface-applied N treatment (*p* < 0.005) (Figure 6).

*Fp* recovery was significantly higher in the *Fp*-inoculated treatment below 32 cm above ground level for LRPB Lancer and below 39 cm above ground level for DBA Lillaroi (Figure 7). A greater separation in *Fp* recovery between the inoculated and uninoculated treatments was observed in LRPB Lancer compared to DBA Lillaroi (Figure 7). Tissue N % of the straw tended to increase in the *Fp*-inoculated treatments and was significantly different in LRPB Lancer between 35 and 47 cm in plant height (Figure 7, left). DBA Lillaroi did not have the same degree of separation in straw tissue N % recovered; however, it did tend to increase in the *Fp*-inoculated treatment reaching significance at approximately 45-47 cm in plant height (Figure 7, right). The lack of correlation between straw N concentration and *Fp* biomass in cereal stems suggests N assimilated by *Fp* was not a driver of tissue N recovery (Figure 7). 

## 3. Materials and Methods

Two experiments were conducted as part of this study. The first experiment was an in vitro study designed to determine the influence of tissue N concentrations on mycelial growth of *Fp* (Section 3.1). The second was a controlled environment experiment where the in-crop effects of different N availability over time on FCR severity were investigated using large field-simulating growth tubes in both inoculated and uninoculated bread and durum wheat plants (Section 3.2).

### 3.1. In Vitro Agar Experiment

This experiment incubated *Fp* cultures on agar plates with various concentrations of N, simulating tissue concentrations of N under low- to high-N regimes. The rate of colony enlargement was measured to determine the interactions between *Fp* and N.

#### 3.1.1. Agar Preparation

Solutions of ammonium nitrate (NH₄NO₃) and urea (CH₄N₂O) were made to concentrations of 0.3 M, 1.5 M, and 3.0 M of each compound. Then, 25 mL of each solution was transferred into 1 L of sterile ¼ PDA + novobiocin agar with agitation supplied by a magnetic stirrer used to mix the compounds evenly through the agar, which had been cooled to 60 °C. The resulting molarity of both urea and ammonium nitrate agar solutions were 0.03, 0.15, and 0.3 M. Using a peristaltic pump, 12.5 mL of each agar treatment was dispensed into sterile 90 mm diameter Petri dishes. 

#### 3.1.2. Isolate Preparation

Three isolates of *Fp* were prepared following methods outlined in [19]. Agar cubes (2 mm^3^) of each isolate were cut from the margin of colonies and transferred onto separate Petri dishes that contained ¼ strength Potato Dextrose Agar (PDA) +  novobiocin (10 g PDA, 15 g technical agar, plus 0.1 g novobiocin/L water). Four replicates of each isolate were cultured under alternating white and near-ultraviolet lights for a 12 h photoperiod of 66.6% alternating fluorescent (FL36W/865, Sylvania, Newhaven, UK) and 33.3% blacklight blue (F36T8 BLB, Crompton lighting, Bradford, UK) light for 7 days at 25 °C.

#### 3.1.3. Plating and Data Collection

Each of the three isolates was plated from a 6 mm diameter agar plug taken from the actively growing outer margin of each prepared culture (see Section 3.1.1). Individual mycelial plugs were inverted in the middle of each plate, then the plate was sealed with clear film and grown in a constant 25 °C room with alternating lighting periods as described previously. Average colony diameter was recorded after 3, 6, and 8 days of growth and converted to colony area assuming circularity. There were 5 replicates of each N treatment and isolate combination.

### 3.2. Controlled Environment Experiment

A controlled environment study was performed with N applied at different depths in the soil profile to alter the timing of N accumulation by the plant. This allowed the temporal measurement of FCR severity relative to N uptake. At senescence, the distribution and concentrations of N in the plant were quantified to determine the Nitrogen Transfer Efficiency (NTE) within the main stem.

#### 3.2.1. Soil, Tube Design, and FCR Treatments

Polyvinyl chloride (PVC) soil tubes, 150 mm diameter × 1200 mm length, were used to simulate a field soil profile (Figure 8). The soil used was a Grey Dermosol with a plant available water holding capacity (PAWC) of 202 mm/m and starting N of 36 mg N/kg nitrate and 4 mg N/kg ammonium. The upper topsoil (top 350 mm) was compacted to a bulk density of 1.1 g cm^−3^ whilst the lower subsoil (bottom 750 mm) was compacted to a bulk density of 1.3 g cm^−3^. Two FCR treatments were used, uninoculated and inoculated. The inoculated treatment contained a band of 20 mm of inoculated soil. This was prepared by evenly mixing ground wheat seed (0.5–2 mm fraction) colonised by 5 different isolates of *Fp* throughout soil at rates of 1 g inoculum/100 g of soil [20]. The uninoculated treatment had 20 mm of soil mixed in a similar manner with grain sterilised in an autoclave (60 min at 123 °C). A further 10 mm of soil was then added to both treatments to minimise colonisation of the fungus across the soil surface during the experiment.

#### 3.2.2. Plant Materials and Growing Conditions

One bread wheat cultivar, LPRB Lancer, and one durum cultivar, DBA Lillaroi, were grown over a six-month period; seeds were treated with Vibrance^®^ (Syngenta, Basel, Switzerland) and Emerge^®^ (Syngenta) at rates of 360 mL/100 kg and 240 mL/100 kg, for standard bunt and smut control and early protection against aphids, respectively. Vibrance^®^ seed dressing also ensured no seedling blight in the presence of *Fp* in inoculation [21]. Six seeds of each cultivar were sown below the inoculum layer, approximately 30 mm below the soil surface, and thinned to four plants per tube upon establishment. There were five replicates of each cultivar and treatment combination. The experiment was conducted in an air-conditioned polyhouse complex at Tamworth Agricultural Institute (TAI), Tamworth, NSW from 6 April to 8 November 2021 with a 25 °C day and external ambient night temperature regime. Soil tubes were individually weighed and watered to field capacity each week until flowering. Post-flowering, the water-stressed treatments were managed to 40% of field capacity (−100 kPa matric potential), whilst the non-stressed treatment maintained the original watering regime of 100% field capacity. Water was administered through a 25 mm PVC pipe located in the soil column consisting of three watering points vertically throughout the profile at 350 mm, 550 mm, and 750 mm below the soil surface (Figure 1). This method was constructed to resemble dryland NGR conditions where summer-dominant rainfall results in minimal in-crop winter rainfall and where plants predominantly rely on stored soil moisture. Three weeks prior to harvest, soil tubes were not watered for a week to dry the soil column down for core removal.

#### 3.2.3. Fertiliser

At planting, soil tubes were treated with basal nutrition of 0.8 g of KCl, equivalent to 50 kg K/ha, evenly mixed in the top 350 mm of soil to rectify K deficiency on basis of mg/kg. The banded N treatment received urea and K^15^NO_3_ in solution at rates of 1.36 g and 0.02 g, respectively, at 350 mm below the surface equivalent to 80 kg N/ha [22]. The surface treatment received the same solution at 50 mm below the soil surface. The rationale for these N treatments was to impose a temporal effect on N availability during the experiment. The surface treatment had N readily available at early growth stages, whilst the banded treatment increased N availability later in the season. The ^15^N treatments provided the ability to observe through tissue sampling when the band was accessed by the bread and durum wheat plants. 

#### 3.2.4. In-Crop Measurements

All plants in each pot were visually scored for the severity of FCR infection based on the extent of browning of stem bases using a 1–3 scale at full flag leaf emergence (GS39), with values presented as a crown rot index [20]. This determined whether all the FCR-inoculated treatments visually displayed signs of infection. The newest fully spooled leaf tip (4 cm) on main stems was taken for N and ^15^N analysis at GS39 and swollen boot (GS45) [23]. Continuous flow isotope ratio mass spectrometry (IRMS) (Sercon 20−22, Sercon Ltd., Crewe, UK) was used to determine N and ^15^N present in the leaf tip at each time of sampling.

#### 3.2.5. Harvest Assessments

Immediately prior to harvest, counts were taken of plants, tillers, and heads. Heads on the main stem from each plant were removed followed by their stems which were first measured for height and then cut 5 mm above the soil surface. The remainder of the heads and stems were then collected separately. Both heads and stems were dried at 40 °C for 72 h prior to weighing. Grain was threshed from the collected heads from the four main stems of plants in each soil tube. Remaining viable heads were collected separately and threshed to recover grain. Visual severity of FCR infection on the main stems was assessed at harvest, as outlined in [6]. For the entirety of the stem at 5 cm intervals starting at the base, the stem was cut, keeping the lower 1 cm for plating of colonisation height by Fp and the upper 4 cm for nutritional analysis. Fungal plating was conducted on ¼ PDA + novobiocin agar and cultured for seven days under the same lighting conditions regime described in the first experiment. Presence of *Fp* was recorded after seven days. The 4 cm nutritional analysis sections were grouped by tube, then trimmed to 5 mm lengths. Near-infrared (NIR) spectroscopy was then conducted on all samples to determine protein levels in grain and N levels in stem residue at 5 cm increments. NIR predictions were validated using a sub-set of tissue samples which were independently assessed using a LECO TruSpec CN analyzer (LECO Corporation, St. Joseph, MI, USA) to accurately quantify N content and apply corrections to NIR predictions. 

### 3.3. Equations 

Nitrogen transfer efficiency was used to determine the effect of FCR inoculation on N efficiency and use within bread and durum wheat plants [24].

Nitrogen Transfer Efficiency (NTE):NTE = YN/YN + TN

Stem Nitrogen Transfer Efficiency (sNTE):sNTE = YN/YN + sTN
where, YN: total N recovered in grain (g), TN: total stored tissue N at harvest (g), sTN: total stored tissue S in main stem at harvest (g).

### 3.4. Statistical Analysis

The statistical software package R [25] was used to fit linear models to the datasets and ANOVA was performed. All factors were included in each model whilst non-significant factors were removed to obtain most parsimonious model. Model diagnostics were checked and, where necessary, data were transformed to uphold model assumptions. Post-hoc multiple comparisons were performed using Tukey’s method (package: lsmeans).

## 4. Discussion

### 4.1. Nitrogen and FCR

It was hypothesised by Davis [13] that there is a relationship between N availability and FCR severity in wheat, where the severity of FCR (measured by visual browning) increased with N fertilisation compared to unfertilised control. The mechanism proposed by Davis [13] was that disease severity (yield penalty) increased due to a secondary effect of the plant consuming early N causing excess vegetative growth and depleting stored soil moisture, which elevated stress at the end of the season. To the best of our knowledge, this is the first study to investigate the primary mechanistic relationship between *Fp* and tissue N to determine the effect on FCR severity in wheat. We suggest that an increase in available N can significantly exacerbate the severity of FCR infection, potentially through increased growth of *Fp*.

### 4.2. In Vitro Growth Relationships

Lower concentrations of urea in agar had a positive effect on the in vitro growth of *Fp*; however, at higher concentrations, urea was found to reduce in vitro growth. These observations support the hypothesis that there is a mechanistic link between *Fp* growth and N availability. A significant inhibitory effect was observed when ammonium nitrate was added to the PDA medium at all concentrations and time points except for one. This effect is likely due to the increased salt load of this compound creating a negative osmotic gradient where *Fp* was restricted in its biological activity, similar results were recorded by [26]. These observations are consistent with a previous *Fusarium culmorum* in vitro growth study, where urea promoted growth while ammonium nitrate inhibited growth. The salt index of urea (1.618), almost half that of ammonium nitrates (3.059), might have allowed mycelial growth below toxicity and provided sufficient N supply to *Fp*. These laboratory observations, although not plant-based, may reflect the interactions between *Fp* and N availability at green plant tissue N concentrations, which the agar N availability simulated. 

### 4.3. Nitrogen Timing Effect on Severity

In the current plant study, FCR severity changed over time relative to soil N availability. Early access to surface N increased FCR severity at the GS39 growth stage. Validation of increased tissue N-contents was observed through NIR analysis, affirming the change in severity was positively associated with relative tissue N concentration in keeping with the findings of the in vitro study. The change in early N recovery was likely not observed directly, because by GS39 the plants had already well-established root systems in both the surface and the band, meaning both treatments had ready access to applied N. However, there was an indication from the ^15^N recovered that there was a difference in early N availability between N treatments, which is consistent with what would be expected comparing banded and surface N treatments. Although N measurements were related to only one harvest timepoint, we believe this response is related to root access to N. As the crop matured and root length density presumably increased deeper in the profile, uptake of soil N shifted from surface to subsoil layers due to water availability. Deeper N recovery is consistent with field observations in the NGR, where low in-crop rainfall maintains dry surface conditions through the growing season [16]. Unfortunately, it appears that the first sampling at GS39 was likely too late to capture early N recovery, as there was no significant difference measured between banded- and surface-applied tissue N concentrations. This shift in the timing of N access or availability resulted in increased FCR severity in the banded N treatments compared to the surface N treatment at harvest, except for *Fp*-inoculated DBA Lillaroi, which continued to increase in the surface treatment. 

A possible explanation for this finding is that from the increased susceptibility of durum varieties to FCR, although its relative N availability changed, the pathogen had adequate N and continued to thrive within the plant. While this observation appears to be at odds with the association of tissue-N concentrations and FCR severity, the scale of the N promotion of FCR may be a subordinate effect to the genetic susceptibility of a variety [27,28]. In the case of DBA Lillaroi, this durum variety may have such a susceptibility to FCR that it becomes heavily colonized by *Fp* irrespective of N status, while in the bread wheat variety LRPB Lancer, relative improvement in resistance allowed N status to influence the severity of FCR infection. This explanation could further be applied to help explain why at harvest FCR severity of DBA Lillaroi, although trending towards favouring the banded-applied N, was only significant under milder infection in the bread wheat variety.

### 4.4. Nitrogen Transfer Efficiency

The need for a global increase in nitrogen use efficiency (NUE) is well established, especially within cereal production systems [11,29,30,31]. Nitrogen transfer efficiency (NTE) considers the efficiency with which wheat plants utilise the N it has captured from the soil for harvested material (grain) and constitutes an important component of overall NUE changes. Fusarium crown rot infection significantly reduced the sNTE of both the bread and durum wheat cultivars in this study by 4.5% and 10.2%, respectively. This resulted in less efficient N transfer to the grain and yet did not appear to be co-located with *Fp* biomass. This study is the first report of an increase in nitrogen uptake due to FCR infection. Importantly, the increase in N use associated with FCR infection did not increase grain protein levels (data not shown). 

Furthermore, when nitrogen placement was assessed, the treatments that typically had the largest differences in FCR severity displayed the largest differences in sNTE. This indicates that as the severity of FCR increases, sNTE is reduced, ultimately decreasing NUE within wheat crops. Therefore, FCR is not only decreasing profitability through a reduction of yield and grain quality but it is also reducing NUE which amplifies the cost of this disease on profitability. At the current time of writing, world fertiliser prices are at record highs, with average urea prices in Australia at approximately 1350 AUD/t [32]. Assuming an average yield loss of 10%, and excluding a quality downgrade, a hypothetical estimation of predicted losses would be approximately 74 AUD/ha [32]. However, if a decrease in NUE of 10% was included, estimated losses to production would be approximately 98 AUD/ha. While these predictions are only estimations based on current prices, it emphasises the importance of the disease, especially concerning its relationship with nitrogen. 

## 5. Conclusions

These findings support the concern about the detrimental impacts of Fusarium crown rot on global wheat production and the related exacerbating effects of nitrogen application on Fusarium crown rot severity. Improved nitrogen management through better nitrogen use efficiency is becoming more essential than ever with increasing fertiliser prices, especially in an environmentally sustainable manner to feed the growing world [11]. The novel finding from this study that Fusarium crown rot infection appears to reduce nitrogen use efficiency presents a considerable additional challenge to improving nitrogen management practices not only in Australia but in other countries where *Fusarium pseudograminearum* is a constraint to wheat production. The controlled environment study demonstrated that banding of nitrogen in water-limited situations may be one method for reducing Fusarium crown rot impacts and improving nitrogen use efficiency by delaying roots interception of nitrogen-rich zones in the soil profile. Additionally, the in vitro *Fusarium pseudograminearum* growth study indicated that further research on N source interactions with Fusarium crown rot severity (i.e., urea vs. ammonium nitrate) appears warranted under field conditions. Current nitrogen management practices when targeting high protein bread and durum wheat production in the northern grains region of Australia, and potentially elsewhere, at/near sowing surface application of urea appear to be exacerbating Fusarium crown rot severity which in turn increases yield loss from this disease. Furthermore, this study highlights that Fusarium crown rot infection also decreases nitrogen use efficiency which is further likely to be exacerbating economic loss from this disease which has been previously unrecognised. This production system, therefore, appears to require greater emphasis on integrated disease management strategies to reduce Fusarium crown rot levels and a re-evaluation of nitrogen management strategies may assist in the management of Fusarium crown rot.

## Figures and Tables

**Figure 1 plants-12-00533-f001:**
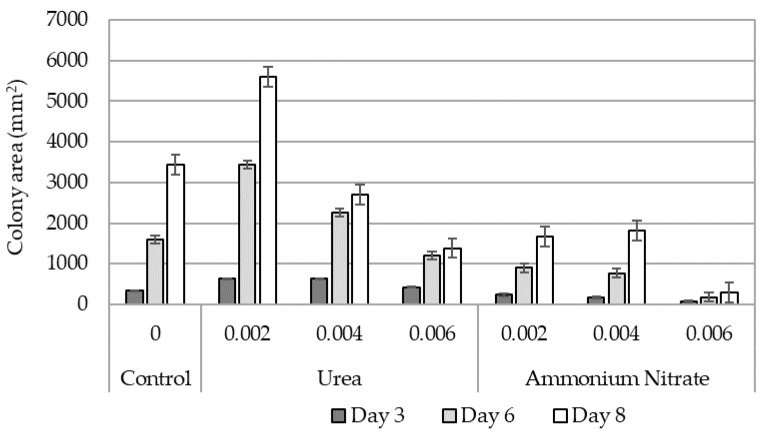
Average colony area (mm^2^) at three, six, and eight days after inoculation with *Fusarium pseudograminearum* isolates. Two nitrogen (N) sources (urea and ammonium nitrate) were compared at equimolar concentrations equivalent to wheat tissue N concentrations (wet) of (0.002, 0.004, and 0.006 mgkg^−1^) against a nil N additive control to determine effect on in vitro growth rate. Error bars indicate standard error. Where error bars do not overlap within each day there was a significant response to added N source.

**Figure 2 plants-12-00533-f002:**
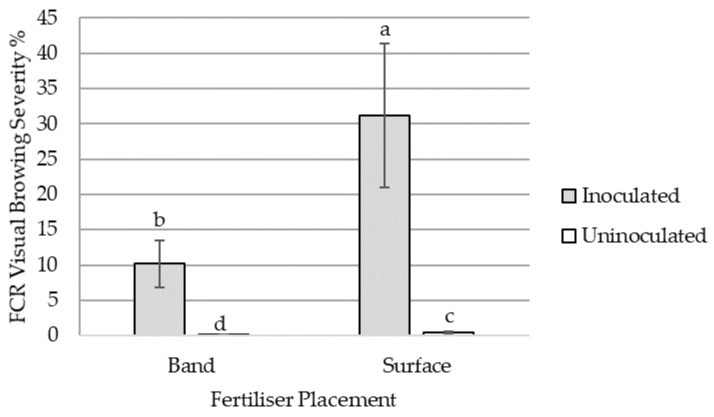
The interaction effects of surface or banded N application on visual FCR severity at GS39 averaged across bread wheat variety, LRPB Lancer, and durum wheat variety, DBA Lillaroi, either inoculated or uninoculated with *Fusarium pseudograminearum*. Standard error bars with different letters indicate significant differences, *p* < 0.05.

**Figure 3 plants-12-00533-f003:**
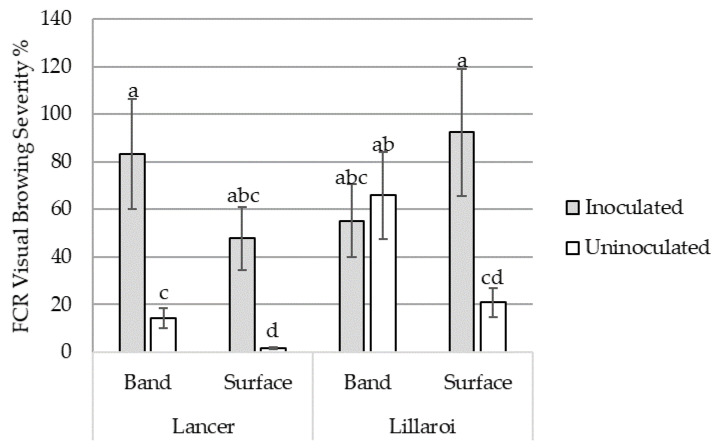
The interaction effects of surface or banded N application and varieties on visual FCR severity at harvest averaged across bread wheat variety, LRPB Lancer, and durum wheat variety, DBA Lillaroi, either inoculated or uninoculated with *Fusarium pseudograminearum*. Standard error bars with different letters indicate significant differences, *p* < 0.05.

**Figure 4 plants-12-00533-f004:**
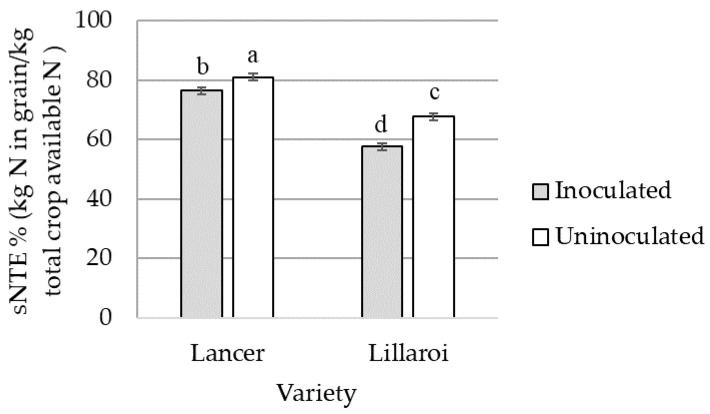
The interaction effects of *Fusarium pseudograminearum* inoculation on stem nitrogen transfer efficiency (sNTE) averaged across bread wheat variety LRPB Lancer and durum wheat variety DBA Lillaroi. Standard error bars with different letters indicate significant differences, *p* < 0.05.

**Figure 5 plants-12-00533-f005:**
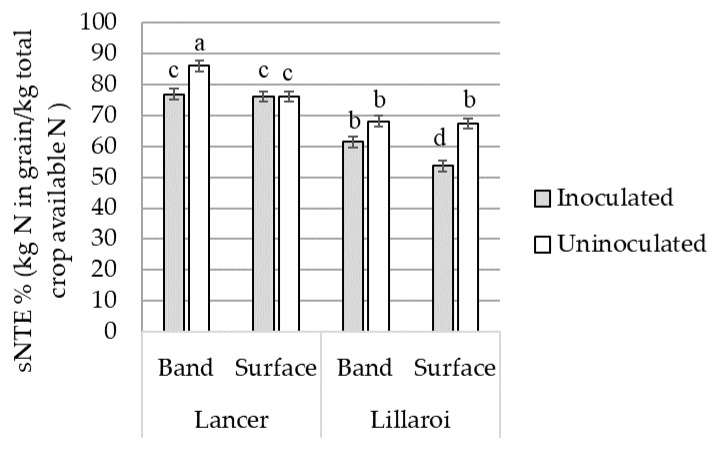
The interaction effects of *Fusarium pseudograminearum* inoculation and N placement on stem nitrogen transfer efficiency averaged across bread wheat variety LRPB Lancer and durum wheat variety DBA Lillaroi. Standard error bars with different letters indicate significant differences, *p* < 0.05.

**Figure 6 plants-12-00533-f006:**
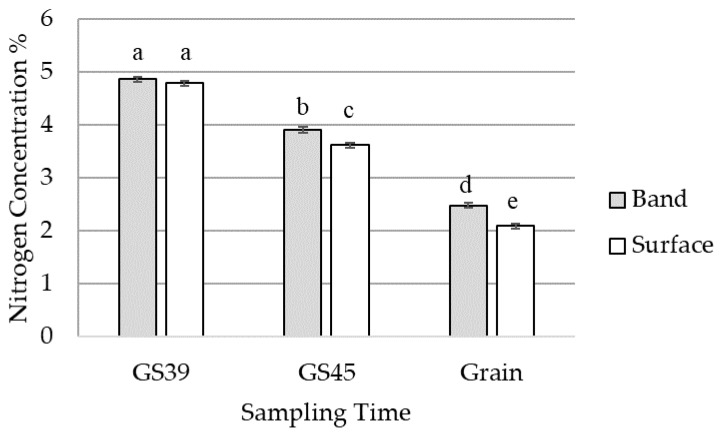
Effect of nitrogen placement on tissue (GS39: emergence of flag leaf, GS45: boots swollen) and grain nitrogen content percentage (%) averaged across bread wheat variety LRPB Lancer and durum wheat variety DBA Lillaroi. Standard error bars with different letters indicate significant differences, *p* < 0.05.

**Figure 7 plants-12-00533-f007:**
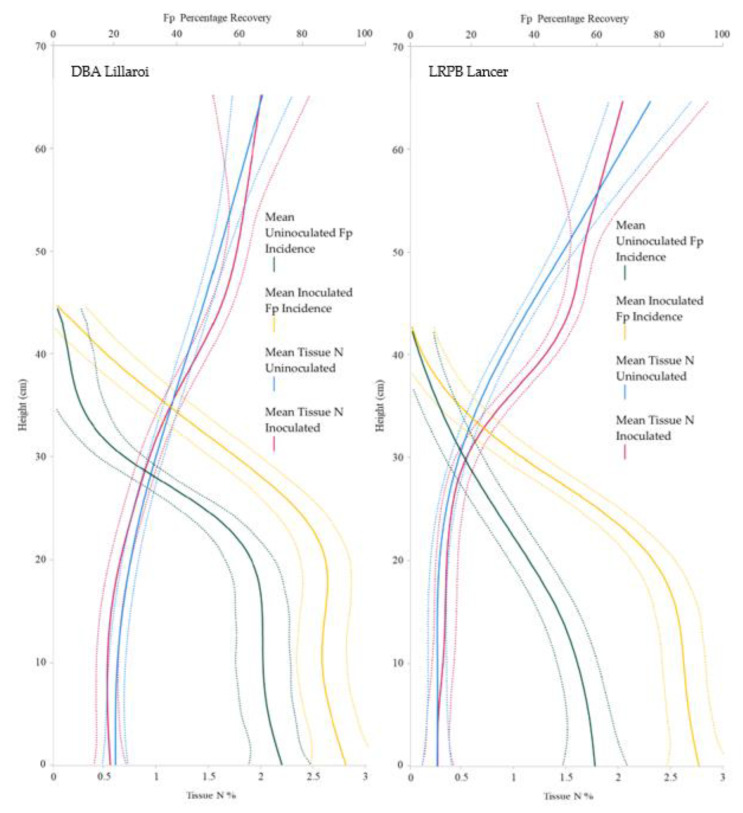
Vertical recovery (main stem only) at harvest of *Fusarium pseudograminearum* incidence and tissue stem N % across bread wheat variety LRPB Lancer (**right**) and durum wheat variety DBA Lillaroi (**left**). Solid line indicates mean with dotted lines indicating upper and lower confidence intervals at 95%, *p* < 0.005.

**Figure 8 plants-12-00533-f008:**
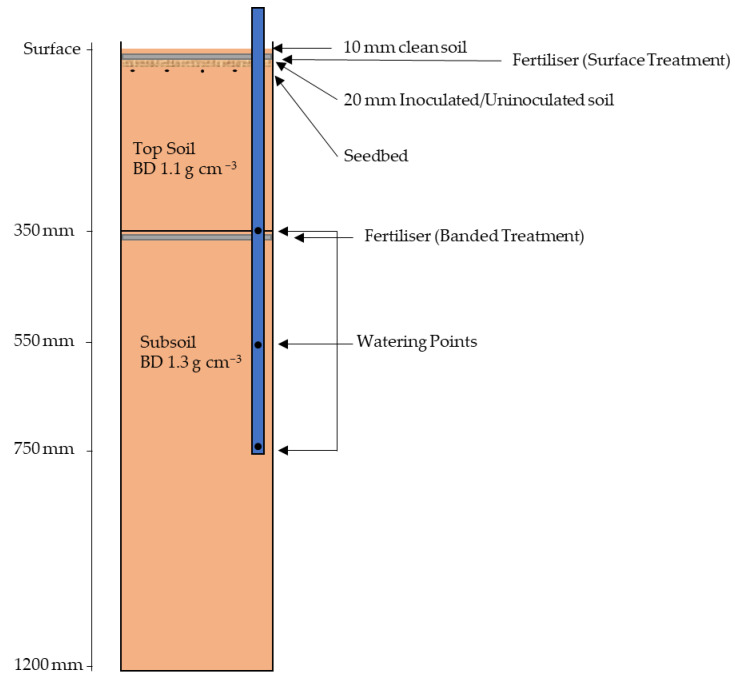
Experimental tube design used in controlled environment study. BD = bulk density.

## Data Availability

Please contact the corresponding author with requests for data.

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
