# Peer review of "Interactions of Fusarium Crown Rot of Wheat with Nitrogen"

_plants, 2023, doi:10.3390/plants12030533_

Round 1

Reviewer 1 Report

The purpose of the article he was reported to understand both the primary mechanistic relationship and the secondary agronomic effects between temporal availability of Nitrogen on Fusarium crown rot severity with a view to identifying potential opportunities to improve management of this disease. Overall, demonstrates that Fusarium crown rot not only decreases yield and grain quality but appears to be having the previously unrecognized detrimental impacts on nitrogen use efficiency in wheat indicating that current losses from Fusarium crown rot are likely considerably underestimated. The article, however, must be improved in terms of writing since some grammar and syntax errors are present in the manuscript. They should address the subject and critically review the information from the literature.

Minor suggestions:

The abstract is written in a way lacks logic. It should highlight the salient findings more critically.

Keywords are present in the title, choose others.

The introduction does say little about the selected plant wheat (Triticum spp.), why these plants were selected? Introduction need more convincing rational for this article.

Provide experimental work plan at the start of M&M. No detail description is available about the experiment.

The results have long paragraphs, I suggest reducing the size of the paragraphs.

The results of this study are not fully explained therefore the interpretation of the results is very difficult. The author needs to provide the % increase or decrease rather than just writing ''significantly increased….''.

Insert in table 3 the standard deviations of the means.

In figures 3, 4, 5, 6 and 7, do the bars in the columns mean standard error or standard deviation?

Authors should discuss the results integrally. The discussion is based on individual results. I suggest that integrating the results will give more value to the work. I suggest that you discuss by integrating all your results. You can use correlation tests (PCA or Pearson Correlation).

Rewrite the conclusion! It needs to be much improved.

Author Response

Minor suggestions:

  1. The abstract is written in a way lacks logic. It should highlight the salient findings more critically.

Authors have amended the abstract to highlight findings more critically.

  1. Keywords are present in the title, choose others.

Authors have amended as per request

  1. The introduction does say little about the selected plant wheat (Triticum spp.), why these plants were selected? Introduction need more convincing rational for this article.

A justification for the importance of FCR to the wheat industry in Australia is already provided in line 34-36.

  1. Provide experimental work plan at the start of M&M. No detail description is available about the experiment.

The authors appreciate the comment and have added generalized descriptive sentences prior to the detailed methods to aid the readership in envisaging the experiment. Lines 68-71 and 110-114 for the two methods respectively.

  1. The results have long paragraphs, I suggest reducing the size of the paragraphs.

 The authors have assessed the size of the paragraphs and reduced the size for ease of reading.

  1. The results of this study are not fully explained therefore the interpretation of the results is very difficult. The author needs to provide the % increase or decrease rather than just writing ''significantly increased….''.

Authors have included reported values as requested wherever possible.

  1. Insert in table 3 the standard deviations of the means.

The error bars in figure 3 are present however due to the scale of the effects, the error bars present on the uninoculated treatments are not highly visible.

  1. In figures 3, 4, 5, 6 and 7, do the bars in the columns mean standard error or standard deviation?

The bars indicate standard error and we have amended the figure captions to clarify.

  1. Authors should discuss the results integrally. The discussion is based on individual results. I suggest that integrating the results will give more value to the work. I suggest that you discuss by integrating all your results. You can use correlation tests (PCA or Pearson Correlation).

We thank the reviewer for this observation but would kindly like to point out that the discussion section already integrates findings from across the results section into this section. For example, the first paragraph of the discussion references both studies contained in this work and integrates figures 2, 4, 7 and 8. We would kindly suggest that this demonstrates an integration of concepts in the discussion and that attempting to correlate findings statically across multiple disconnected experiments would unnecessarily complicate the story of the paper.

  1. Rewrite the conclusion! It needs to be much improved

Thank you for your comment, we have re-considered the conclusion and agree with reviewer 1 that the conclusion summarises the findings of the work.

Reviewer 2 Report

The topic is original and falls within the scope of the journal. Overall, the manuscript is well addressed, including the language.

Introduction is appropriately provided.

Keywords must be improved avoiding the repetition of words in the title.

M&M is overall well described. However, some issues are found, as follows:

-Authors should indicate how many plants per tube were sown and how many plants per tube were used to test. 

-Authors should also verify the greenhouse experimental period.

-Indicate how many plants were visually scored?

-L142: Please define the phases "GS39" and "GS45"

-Authors must indicate references for equations (NTE, sNTE)

-Authors should describe the 15N experiment (fertilizer used, fertilization, method used, references)

Results:

-L177: where can we find the data? Section 3.1 is not clear. Please improve.

-Fig. 2: authors should indicate how they analyzed the data. Is this the result of an interaction?

-Figs. 3, 4, 5, 6: add "The interaction effects of..."

-Fig. 4: add in XX the legend "Varieties"

-Fig. 7: add in the legend: "GS39"=..., "GS45"=...

-Fig. 8: add the name of varieties in XX legend

-L202/203: 15N data are not shown because no significant response was obtained. It woul be interesting if authors present as a Suplemmentary Material. 

-L211: add "(Fig. 4)"

-L212/214: "...except for the durum...(P<0.0001).": This is not correct. All of them are statistically equal.

-L222: add "(Fig.5)"

-L240: replace "content" by "concentration"

-L262: write: "...by Davis (11)..."

-L264: write: "control"

-L269: write: "suggested"

-L291: write: "concentration"

-L319: "NUE": spell

Conclusions are overall well addressed but authors should avoid the use of abbreviations, including "Fp".

Round 2

Reviewer 2 Report

Authors improved the manuscript according to overall Reviewers’ recommendations. However, some issues are still found in the present manuscript, as follows:

-In the Introduction, authors should describe the importance for choosing the wheat crop.

-Authors shoud indicate references for equations (NTE, sNTE).

-L353: Please write “(NUE)”

- In the Discussion, authors should integrate the discussing results.

-Authors should remove the abbreviations in Conclusions section (e.g., FCR, NUE)

Author Response

Response to Reviewers

Comments and Suggestions for Authors

Authors improved the manuscript according to overall Reviewers’ recommendations. However, some issues are still found in the present manuscript, as follows:

  1. In the Introduction, authors should describe the importance for choosing the wheat crop.

Amended as requested:

L33-34: “Wheat is the world’s most important food source, providing 19% of the daily calories and 21% of protein requirements for humans [1, 2].”

  1. Authors should indicate references for equations (NTE, sNTE).

Supporting reference [24] added L195

  1. L353: Please write “(NUE)”

Amended as requested

L355: “nitrogen use efficiency (NUE)”

  1. In the Discussion, authors should integrate the discussing results.

We acknowledge this observation by the reviewer but feel that the discussion section already adequately integrates findings from across the results section. For example, the first paragraph of the discussion references both studies contained in this work with the proposed mechanism summarised on L305-306. The findings although separated into subsections in the discussion, is we feel also adequately integrated within the conclusion section. Happy to amend further if requested with some further guidance (e.g. examples) from reviewer if still concerned.

  1. Authors should remove the abbreviations in Conclusions section (e.g., FCR, NUE)

Amended as per requested L378-402